# Re-Thinking the "Problem" in Inquiry-Based Pedagogies through Exemplarity and World-Oriented

**Simon Warren**

Centre for Research on Problem-oriented Project Learning, Roskilde University, 4000 Roskilde, Denmark;
warren@ruc.dk

**Abstract:** This paper conducts a theoretical exploration of the inquiry-problem in problem-oriented pedagogies. Specifically, the article draws on a critical reflection of the addition of a global and internationalisation dimension to the problem-oriented project learning (PPL) pedagogic model at Roskilde University in Denmark. While the tradition of PPL has always promised a world-oriented and transformative alternative to traditional higher education, the article argues that this new global dimension presents an opportunity to renew the transformative potential of PPL. In particular, it argues that it can facilitate new ways of conceptualising the inquiry-problem in relation to the pedagogic idea of exemplary problems. Furthermore, problem-oriented approaches can generally be articulated with a more values-based conception of internationalisation and global justice, in order to enhance the transformative potential of these pedagogies. The article proposes that this enhanced conceptualization of world-orientation is an appropriate answer to the call for pedagogic responses to the existential threat posed by the climate crisis.

**Keywords:** problem-oriented project learning; problem-orientation; exemplarity; PBL; global justice; internationalization; higher education

## 1. Introduction

This paper reports on a theoretical exploration conducted by the author of the inquiry-problem in problem-oriented pedagogies, through the specific instance of how exemplary problems are conceptualised in problem-oriented project learning at Roskilde University (RUC) in Denmark. Specifically, the nature of the inquiry-problem is examined in relation to the concepts of problem-orientation and exemplarity, tracing the intellectual origins of these concepts and their development as pedagogic practices at RUC. The pedagogic concept of exemplarity may not be familiar in Anglo-American educational discourse. Apart from the example of Aalborg University in Denmark, exemplarity is not a central concept within debates on problem-based learning [1]. The concept of exemplarity is defined in Section 2 and expanded on further in Section 5. The particular form that inquiry-problems take at RUC is rearticulated through the concept of world-orientation, drawing on both the original theoretical foundations of RUC's pedagogic model and more recent discussions of world-oriented education. This is the basis on which the paper seeks to contribute to re-thinking the inquiry-problem in problem-oriented pedagogies. By implication, this theoretical exploration raises questions about the purpose of higher education in the context of the existential threat posed by the climate crisis.

*Problem-Oriented Pedagogies for a Dangerous World*

The challenges facing the world, and the place of humans in the world, are daunting. Climate change, resource depletion, food and water insecurity, social and political upheaval, migration, and violent conflict confront us on a daily basis. The traditional university curriculum, based as it is on disciplinary fields, seems at odds with this reality. Let us take just one example, that of the melting glaciers

on Greenland, something that has come to symbolise the imminence of climate change. To fully grasp the phenomenon of the melting glaciers on Greenland requires, at the very least, an interdisciplinary approach. It requires expertise from a range of related disciplines including climatology, geophysics, marine science, oceanography, botany, and many others [2]. It is clear how these might relate to each other around the inquiry-problem of melting glaciers in Greenland. The recent phenomenon of migration to Europe from regions such as Syria or Afghanistan and Libya might suggest a similar interdisciplinary approach, that could encompass political science, sociology, human geography, and security studies. However, this phenomenon also calls for a transdisciplinary approach, one that transcends the social and human sciences on one hand, and natural sciences on the other. While the social and political basis of these migration flows are evident, they are also intimately related to climate change that affects water and food security, and contributes directly to social and political upheaval. Yet, when a student enters higher education, they will typically encounter a curriculum that is organised around discrete disciplinary fields with limited interaction across these fields, particularly at the undergraduate level. This brief discussion serves to illustrate the disconnect between the traditional university curriculum and the experience of real-world problems we face. It is obviously important that specialist expertise is developed. This is not an argument against specialisation. It is, however, an argument for the role that problem-oriented pedagogies can play in building deeper and broader understandings of how the world works [3,4].

There are, however, traditions and developments within higher education that place inquiry and problem-orientation at the centre of their pedagogies. In the 1970s, there emerged a cluster of curriculum developments that can broadly be termed problem-based or problem-oriented [5,6]. It is difficult to define these with any precision since they have taken on a variety of constellations [7] that configure knowledge, student participation, the role of the teacher, methodology, and the origin of the inquiry-problem in diverse ways. An early review of the relevant literature [8] proposes that these problem-oriented pedagogies stressed real-world relevance in three main ways—development of skills relevant for professional practice; development of action or problem-solving dispositions relevant to the real world, particularly in relation to the world of work; and the fostering of independent inquiry. While this is a rather narrow and limited definition of action-in-the-world, it does indicate that a world-oriented approach is critical for problem-oriented and problem-based pedagogies. The emergence of this approach in the 1970s is of importance in relation to the main focus of this paper and will be discussed more fully later. There is, of course, a longer tradition of problem-oriented pedagogies associated with the work of John Dewey. Although his work has, mistakenly, been associated with certain forms of experiential learning, his conception of experience articulates a relationship to the world and education as a reflexive inquiry in relation to the world [9–11]. His ideas about experience and student interest will be taken up later. Of importance at this point is that this tradition constructs the inquiry-problem as one that is world-oriented both in its content but also in its desired outcome in terms of transformed action-in-the-world. This construction of the inquiry-problem takes us beyond the limited conception of action-in-the-world outlined above. If, as Dewey suggests, education is inquiry, and inquiry requires an inquiry-problem, what is an educative inquiry-problem? There is a broad debate on this in relation to Dewey. According to Luntley [12], an inquiry-problem begins with an unsettling feeling about our relation to the world. Education as inquiry, in this Deweyan sense, does not lead to simple problem-solving but to a pedagogy that further unsettles the student through an extension of experience and concepts and, therefore, a transformed relation to the world. The inquiry-problem that education is therefore oriented towards is thus more than skills, problem-solving, or independent learning, but invites a transformed relation to the world and, hence, action-in-the-world. Such an orientation seems relevant and necessary in the context of contemporary existential challenges and threats.

The 1970s saw, in Denmark, the formation of two institutions of higher education that were deliberately problem-oriented in terms of pedagogy and curriculum. Whilst Aalborg University has been closely associated with problem-based learning [6,13], RUC has developed an inter-disciplinary problem-oriented pedagogy and curriculum. The Deweyan way of conceptualising the inquiry-problem resonates with RUC's pedagogic model, which forms the empirical focus for this paper. As will be

argued below, the presentation of the inquiry-problem as world-oriented has been a feature at RUC since is foundation. However, it is proposed that the introduction of a new institutional commitment to a global dimension and internationalisation can be viewed as an invitation to revisit the way the inquiry-problem is conceptualised in relation to the existential issues facing us. The paper begins by briefly describing the emergence of RUC as a social experiment in higher education and the centrality of the exemplary problem for its pedagogic model and curriculum. RUC is then situated within the contemporary European policy-scape. This begins with a description of RUC's commitment to internationalisation. This section proposes that the European response to diversity and migration offers indications of how this commitment to internationalisation can be translated pedagogically, particularly in relation to the inquiry-problem. The paper then further develops the concept of world-orientation as an option for translating internationalisation into pedagogy. Specifically, this concept is used to re-articulate RUC's approach to the inquiry-problem appropriate to the existential issues. The paper presents an indicative idea of what this would look like in the context of problem-oriented project work. Finally, the paper highlights the wider implications of this theoretical and pedagogical exploration for problem-oriented pedagogies. In particular, the paper suggests that this approach to conceptualising the inquiry-problem responds to increasing calls for an adequate pedagogic response to the climate crisis.

## 2. Roskilde University: A Social Experiment in Higher Education

The 1960s and 1970s saw a number of critiques of what were perceived to be elite institutions of higher education and increased demand for post-secondary and further education. This included calls for an expansion of higher education opportunities and questions about the relevance of the curriculum to the modern, technological world. In the UK, for instance, this saw not only the development of new universities organised around innovative curricula and pedagogy, but also vocationally oriented polytechnics [14]. In Denmark, there were similar calls for a more democratic orientation to higher education and for socially and economically relevant curricula [15,16]. This saw the establishment of two new institutions of higher education in Aalborg in the north of Jutland and Roskilde to the west of Copenhagen. In this sense, both institutions reflected the purposes of problem-oriented learning outlined earlier, of forms of higher education focusing on society (in the form of professional training or social relevance), and innovative curricula and pedagogy.

RUC was established in 1972 as a social experiment in higher education [17]. Born out of the social and political movements of the late 1960s, RUC aimed at developing a form of higher education that was oriented towards a critical social engagement with the world. Although RUC was established under a conservative government, it was influenced by demands from radical left students and academics for a different kind of pedagogy that challenged traditional hierarchies of knowledge and power in universities [18]. The choice of a problem-oriented and interdisciplinary approach was heavily influenced by the student movement. It is also important to point out that, in its original formulation, education at RUC was structured within a five to six year master's programme typical of all Danish university education, rather than the Anglo-Saxon undergraduate model. The original model was organised around project groups with concomitant thematic courses that supported an interdisciplinary approach. The emerging model involved a more intense relationship between students and lecturers, with students having a significant role in directing the project work. Over time, the governance of education at RUC would change due to external pressures. Today, an undergraduate structure has replaced the previous master's structure. Although the interdisciplinary nature of the education programme has been muted over the years, specialisation is still deferred and subject combinations enable some degree of interdisciplinarity. The RUC model has found form in a pedagogic approach that is currently known as Problem-oriented Project Learning (PPL) and codified in its 7 Principles (https://ruc.dk/en/problemoriented-project-learning-pedagogical-model-roskilde-university): Project-work, Problem-orientation, Interdisciplinarity, Participant direction, Exemplarity, Group-work, and International insight and vision.

There is not space in this paper to adequately describe each of the principles and the relevant theoretical influences. A description of a common undergraduate programme will help to illustrate the

basic educational structure at RUC. This is followed by a brief review of the underlying theoretical ideas. In a common undergraduate programme, a student will begin each semester by forming a project group with a number of other students around a common inquiry-problem. They will negotiate the specific nature of this problem with a supervisor and devise a research strategy that includes relevant theories and methodologies, drawing on a range of related disciplinary fields. Over the semester, as well as in supervision meetings, the project group will be required to present their work-in-progress in a number of evaluation events, where other students and supervisors will have the opportunity to peer-review the project. The project group will then produce a project report (which could be a product, technical solution, etc., depending on the disciplinary fields and problem) and participate in an oral examination resembling a public PhD defense. Parallel to this process of inquiry are courses on philosophy of science, methodology, and disciplinary subjects. This is repeated every semester, with project group-work constituting 50% of the curriculum. The significance of the project work distinguishes PPL from both more traditional curricula and other forms of problem-based learning (PBL).

Underpinning these principles lies a composite pedagogy [19]. Based largely on the theories of Danish educationalist Knud Illeris, PPL foregrounds an experiential and problem-focused educational process. The educational process is understood as needing to transform the way a student understands the world and their relationship with the world. The starting point is seen to be the students' interests and, therefore, their prior experience of the world. But the educational process needs to transform that experience, expanding the student's knowledge and affective dispositions (problem-orientation). The ultimate objective is for education to result in free and socially critical individuals, though Illeris also noted the employment relevance of developing independent learners who can work collaboratively and problem-solve. Although the problems that students work on need to be authentic to the students themselves, they also need to be transformative in the way they orient the student to the world. Problems need to be connected to larger social issues or knowledge problems (exemplarity). In its original articulation, this drew on the work of the German sociologist Oskar Negt, where exemplary learning could lead to radical action aimed at human liberation. Negt drew on C. Wright Mills' work on the sociological imagination [20] and, specifically, the relationship between personal troubles and public issues, between subjective experience and historically constituted social structure. Negt wrote his book *Soziologische Phantasie und exemplarisches Lernen* (Sociological Imagination and Exemplary Learning) as a contribution to critical trade union education and work-based learning. The objective of this theoretical contribution was to support a form of education that enabled workers to analyse societal conditions in the context of their subjective experience, and therefore understand the basis for these subjective experiences, and identify ways of rectifying these conditions [21]. Exemplarity, then, in the context of problem-oriented project work, asks students to view the particular inquiry-problem as exemplifying a larger social issue.

Exemplarity as the relation between the particular and the general can also refer to an epistemological problem within a field of study, where the students inquire into a knowledge problem as exemplary of the disciplinary field. This draws on the German educationalist Martin Wagenschein. Oriented towards epistemic problems in the way that a knowledge problem connects with larger categories of knowledge, the inquiry-problem must throw light on key questions in the discipline, and the particular focus for inquiry should open up the discipline, making it more accessible to the student. Thus, the idea that the learning of a particular phenomenon can engage the learner in understanding a complex totality was reinterpreted in the same sense by which a particular socio-political event can critically reflect a political totality [22] (p. 77). Patronis et al. argue that epistemological problems need not be clear cut but can be a contested area of knowledge and thus provoke debate. They argue that this leads towards a more hermeneutic or interpretive conceptualisation of knowledge problems [23].

Education does not just involve learning about the world but changing one's action in the world. In this brief outline of PPL as a composite pedagogy, the principles of problem-orientation and exemplarity have been highlighted. This is not to suggest that the other principles are of less significance. PPL as a practice should be understood as involving the relationship between the principles rather than viewing

them as separate items, but certain principles are more foundational than others. The interdisciplinary nature of any PPL inquiry arises from the problem (problem-formulation as an exemplary problem) that is inquired into. Students work collaboratively, acting as peer-reviewers in collaboration with supervisors (group work). Whilst the work of Andersen et al. has sought to describe the emergence of the pedagogic model at RUC and its contemporary practices, this paper re-articulates the core concepts of the RUC model. Specifically, the paper aims to expand the world-oriented nature of the pedagogic model.

Highlighting problem-orientation and exemplarity also draws attention to the distinction between PPL and other forms of PBL. Perhaps most important is the different nature of the inquiry-problem and who generates the inquiry-problem. Drawing on the work of Helen Walkington [24], who has studied undergraduate students as researchers, RUC students, in the context of the project work that constitutes 50% of the curriculum, engage in inquiry projects similar to classical research. Unlike most PBL, where the problem is often directed by academic staff and where there is little scope for practical inquiry over a sustained period (a whole semester in the case of RUC), in PPL, inquiry-problems are mostly student initiated and directed in consultation with academic supervisors and reviewed by peers and academics.

The introduction of Principle 7, on international insight and vision, as will be discussed below, has not entered as a primarily pedagogic principle. Therefore, it requires further translation into pedagogy. This paper contends that this principle can strengthen the transformative potential of the inquiry-problem in PPL. If this is the case, then this international and global perspective could have transformative implications for all forms of problem-oriented pedagogies.

## 3. International Insight and Vision in the European Higher Education Policy-Scape

Although the presentation of the RUC model in terms of a set of pedagogic principles is relatively new, these are principles that have animated the pedagogic approach since its foundation. The introduction of the principle of international insight and vision is a novel addition. The relationship between the language of this principle and wider discourses of higher education's internationalisation will be discussed later. For the moment, it is sufficient to describe the principle and how it emerges in a European policy context.

The principle of international insight and vision sets an ambitious challenge for education programmes at RUC and desired student outcomes,

> Roskilde University attaches great importance to all education programmes having an international orientation. Integration of international, intercultural and global aspects is central to the organisation and implementation of research-based teaching and education, and is strongly based on research's cultural and historically embedded international vision. Education programmes with an international perspective enable students to engage in a complex everyday life in an internationally oriented labour market and strengthens their ability to understand and act on global issues. Through the academic material, networks and meetings with international researchers and fellow students, the students acquire perspectives on issues that cut across disciplines, national borders, culture, language and nationality. In addition, the students develop intercultural competencies alongside refining their ability to identify, analyse and reflect on global challenges and national and regional differences. The goal is thus to develop global awareness and citizenship, intercultural understanding and communication, critical engagement, as well as tolerance and respect through knowledge and insight. [25]

The language of the principle contains a combination of values-based educational ambition (global awareness and citizenship), as well as labour market-oriented outcomes (intercultural competencies and problem-solving abilities). This combination is echoed in European higher education policy, in particular in the dual role of higher education in securing economic growth and European competitive advantage, and in social inclusion and cohesion. Prior to 2015, European higher education policy has been focused on the convergence of higher education systems, with the objective of making Europe the

leading knowledge economy in the world [26,27]. The mobility of students, academics and knowledge was projected as necessary to this objective, leading to the provisions of the Bologna and Lisbon processes [28–30].

The 2015 migration crisis and the rise of xenophobic and nationalist movements across Europe provoked a number of responses from the EU in relation to the European Higher Education Area (EHEA). The Yerevan Communiqué made a call for an inclusive response to migration to be emphasised in the context of the Bologna and Lisbon processes. In particular, it called for the development of inclusive environments in institutions of higher education. The Communiqué seeks to reiterate the commitment to the Bologna process but is clearly a response to the 2008 financial and economic crisis and consequent austerity policies, the intensification of migration to Europe as a consequence of conflicts in Africa, the Middle East, and Afghanistan, and the perceived threat of political Islam and the rise of nationalist and xenophobic movements in Europe,

> Today, the EHEA faces serious challenges. It is confronted with a continuing economic and social crisis, dramatic levels of unemployment, increasing marginalization of young people, demographic changes, new migration patterns, and conflicts within and between countries, as well as extremism and radicalization. [31]

As well as advancing inclusion through employability, the Communiqué stresses the need to enhance the quality of teaching and learning in higher education and the building of more inclusive higher education systems. This objective of inclusion is connected with the need to improve access to, and effective participation in, higher education. The Communiqué continues to stress the mobility commitments of the Bologna process, rearticulating this in terms of ensuring that those from diverse and/or economically disadvantaged backgrounds can benefit from such mobility.

These commitments were further enhanced in 2017 in the renewed EU agenda for higher education [32], which highlighted the need to build environments within higher education that allowed a diversity of students to succeed. This linked the modernisation of higher education to the existential threat to the 'European project' posed by Brexit—the rise of political movements in Europe sceptical about EU harmonisation and articulating nationalist and xenophobic sentiments. While stressing the economic rationale for modernisation, the renewed agenda for higher education does connect teaching and learning explicitly to the objective of inclusion,

> Making higher education systems inclusive also requires having the right conditions for students of different backgrounds to succeed. This goes beyond the question of financial support for disadvantaged groups, although this is vital for those from low-income backgrounds. To promote the successful completion of studies, higher education providers should take a holistic look at how teaching and assessments are organised, put measures in place to mentor students, and provide academic and non-academic support.

European higher education has been given the responsibility to respond actively to Europe's social and demographic challenges, and teaching and learning are key strategic areas for the development of inclusive higher education systems. This key objective is linked to the economic rationale of addressing a perceived mismatch between the high skills required for a competitive European economy and the stock of human capital amongst Europe's young people and securing the European project itself.

In 2018, the European University Association produced a report entitled 'Universities' Strategies and Approaches towards Diversity, Equity and Inclusion: Examples from across Europe' [33]. In this report, it was noted that, despite commitments to inclusion in higher education at both the European and international levels, "only a few countries have followed up with concrete action at the system level to foster social inclusion in higher education" [33] (p. 4). The EUA report focuses on examples of responses by national higher education systems as well as selected institutional responses in Ireland, France, Germany, Netherlands, Catalunya, Austria, and England. The report illustrates the diverse range of strategies adopted and issues that are prioritised, particularly at the institutional level. Recent

comparative research into constructions of diversity and inclusion in European higher education policy also notes the broad range of strategies and priority areas highlighted above [34]. None focused on the mainstream curriculum. The recent 'Trends 2018' report by the EUA [35] provides more evidence of this lack of focus on the mainstream curriculum. Although social inclusion was a topic considered by the majority of institutions to some degree, there was limited impact on the mainstream curriculum. Only 28% of institutions stated that social inclusion was considered in normal learning and teaching practice. The majority of practices relied upon the provision of additional and group-specific supports.

RUC's principle 7 appears to go beyond these limited responses, or at least has the potential to do so. This principle of international insight and vision, focused on the mainstream curriculum and requiring all education programmes to incorporate an international perspective, is more in line with other strands within European higher education policy. Sue Robson and Monne Wihlborg [36] highlight the importance of the European Commission communication entitled 'European Higher Education in the World' in terms of its more comprehensive articulation of internationalisation, oriented towards cultivating the value of global citizenship. This articulation of internationalisation resonates with RUC's principle of international insight and vision. The critical question explored in this paper is whether the potential of this principle can be realised in relation to the nature of inquiry-problems in this particular problem-oriented pedagogy.

## 4. Towards a Values-Based Global Justice Perspective

This complex policy-scape poses empirical and normative challenges for higher education. Empirically, it forces all actors within higher education to reflect on how globalisation, diversity, and pressures for international performativity transform the purposes and practices of higher education institutions, and how the strategic and practical actions of higher education institutions transform the wider environment. Normatively, it provokes consideration of how we should respond to such transformations [37]. One response to these challenges is to adopt internationalisation strategies.

Internationalisation, as a specific object of higher education policy and strategy, is a relatively recent phenomena [38]. In Europe, this has been given further stimulus since the 1980s, with the introduction of the Erasmus programme, Bologna, and the Lisbon agenda. Internationalisation was initially characterised by cross-border movement of students and academics in line with European Commission commitments to the mobility of knowledge. However, due to the increasing influence of market and competitive dynamics in global higher education [39,40], internationalisation strategies, at both the national and institutional levels, have emphasised the export of education programmes (branch campuses, franchised education programmes) [41,42], competition for high-performing postgraduate students and academics to increase system and institutional status, and research, rather than teaching collaboration [43]. Alongside these trends, there has been a growing recognition that most students and academic staff are not mobile. Consequently, there has been a conceptual and practical shift to consideration of internationalisation at home [44], the international campus, and internationalisation of the curriculum [45]. This understanding has coalesced around a normative definition of higher education internationalisation as the "the process of integrating an international, intercultural, and global dimension into the purpose, functions (teaching, research, and service) . . . " [46] (p. 11). Indeed, this recognition lies behind the specific articulation of an internationalisation strategy by the Danish government that called for the creation of international learning environments to replicate the kinds of intercultural experience and outlook presumed by student mobility [47]. It is not surprising, then, to see this formulation echoed in RUC's Principle 7. The more detailed institutional description of this principle states that:

> Roskilde University attaches great importance to all education programmes having an international orientation. Integration of international, intercultural and global aspects is central to the organisation and implementation of research-based teaching and education, and is strongly based on research's cultural and historically embedded international vision. [25]

This institutional definition also echoes a values-based conception of internationalisation. This values-based aspect is captured in the desired student outcomes, where the principle seeks to develop student dispositions of intercultural and global awareness, global citizenship, tolerance and respect for difference.

A number of scholars have highlighted the increasing instrumentalisation of internationalisation strategies, particularly in relation to the status economy of higher education. Institutional internationalisation policies are often driven by a number of instrumental concerns. Universities are asked to produce graduates who can contribute to and participate in competitive globalised economies. Internationalisation, therefore, is concerned with cultivating certain kinds of global knowledge and intercultural competencies related to economic activity. In Europe, policies are also entwined in the EU commitment to the mobility of knowledge, students and academics, and the convergence of higher education governance. At a global scale, this is further articulated with research competitiveness mediated by university league tables and journal impact factors, driving institutions to attract high-performing scholars and students to improve their marginal position in various rankings [48–50]. For instance, Knight [51] (p. 84) notes that "Capacity building through international cooperation projects is being replaced by status building initiatives to gain world class recognition and higher rankings". Robson and Wihlborg [36] (p. 128) contrast this instrumentalisation with a values-based approach to internationalisation, as a "values-based movement that improves the quality of teaching, learning and research, enhances the experience and understandings of staff and students, and addresses societal issues to improve cross-cultural understanding, inclusion and social justice".

If the values-based potential of RUC's principle 7 is to be realized, then we need to clarify what values pedagogic practices will be based upon.

Principle 7 proposes that,

- Education programmes with an international perspective enable students to engage in a complex everyday life in an internationally oriented labour market and strengthen their ability to understand and act on global issues;
- Through the academic material, networks and meetings with international researchers and fellow students, the students acquire perspectives on issues that cut across disciplines, national borders, culture, language and nationality;
- In addition, the students develop intercultural competencies alongside refining their ability to identify, analyse and reflect on global challenges and national and regional differences.

Such an approach would appear to address the concern about the economically instrumental orientation of much of the higher education internationalisation strategy. However, all of these can be pursued without breaking with this instrumentalization of internationalisation. Using the relationship between the global North and South as a critical example, we can re-articulate RUC's principle 7 objectives as negatives [52–54],

1. Internationalisation can focus on intercultural understanding in purely instrumental terms related to global employability, without really questioning the historically formed inequalities between the developed North and undeveloped South;
2. Understanding the interrelatedness between North and South, including negative impacts does not necessarily translate into an understanding of how Northern dominance, and our ways of life, require the South's subjection—the South is a place to know about, to apply our methodologies to, and extract wealth from;
3. And, addressing the existential threats before us may be approached whilst holding onto the growth logic that underpins Northern economic and political dominance, and which also produces internal inequalities.

Scholars and practitioners working in the field of global citizenship education have tried to articulate what an ethical or values-based internationalisation might look like, arguing for the values of

global justice and equity as an organising frame. Lynette Shultz [4,52] proposes a pedagogical approach that requires that educational inquiry should recognise the complexity of any issue, analyse an issue through a historical, geographical, and epistemological lens, and, finally, engage with a plurality of world views, knowledges, and experiences (particularly those most affected by the issues). The aim of such a social justice values-based approach is not to replace one set of content with another, but to activate a dialogical relationship between students, educators, curriculum content, and world issues.

We have here some indications of what a values-based internationalisation might entail. The argument here is that this formulation of values resonates closely with the world-oriented and transformational intent of PPL. In the following section, this values-based approach will be further examined in relation to the core concepts of problem-orientation and exemplarity.

## 5. Ethical Internationalisation, Problem-Orientation, and Exemplarity

The pedagogic approach advocated by Shultz involves an ethical and political commitment to the world and others in the world, a willingness to interrogate the geopolitical foundations of knowledge and knowledge production, and an incentive to act in the world based on these commitments. This global orientation can both articulate the pedagogical principles of problem-orientation and exemplarity, and re-articulate them as powerful forms of transformative education. Practically, within RUC education programmes, these two principles are particularly enacted in a process termed problem formation. Problem formation occurs at the beginning of a semester where the project group are required to construct a problem-oriented project, but also develop it as an exemplary problem. This begins with the initial subjective interests of each student. For Dewey, this is an important pre-requisite. Inquiry-problems must connect authentically to the student in order to arouse intellectual curiosity beyond the instrumental concerns of the curriculum [9]. In the context of project groups, this also necessitates negotiations towards consolidating a group interest. But consolidation of a group interest, in this sense, should, according to Dewey, propel students beyond what is known or comfortable [10]. Similar to the idea of threshold concepts, a project should be challenging and may produce a sense of liminality in order to expand their cognitive horizons [10,55,56]. An inquiry-problem should therefore be something that is uncertain and problematic, requiring students to go beyond existing habits of thought. However, for Dewey, education is not just about the development of new understanding and perceptions, or the integration of new knowledge. It should contain the possibility of enriching a person's being in the world, resulting in a changed orientation to the world [9]. Similarly, Masschelein and Simons [57], in their discussion of school education, have proposed that the purpose of education is 'awakening interest' in a world beyond the individual—an interest in others and the world we inhabit. Morten Korsgaard (2019) has developed this further with his concept of self-outsight. For Korsgaard, exemplarity is about exciting interest in the world we inhabit beyond ourselves. Of critical importance for Korsgaard is that we do not live in an individual world, but a world of plurality,

> Plurality does not appear coincidentally and through the emergence of the individual. It arises as the consequence of human beings gathering collectively around the things that interest them, and of the renewal of the human habitat through actions that aim to make a mark on it and leave it in a better state for generations to come. [58] (p. 165)

This world-oriented perspective is also argued for by Gert Biesta [59].

This world-oriented conception of education as an integral part of problem-oriented pedagogies can be further enhanced through an articulation of Shultz's values-based approach. We can see how this might stimulate us to think again about problem-orientation and exemplarity through a consideration of how the problem is defined, whose interests are served by the way the problem is defined, what knowledge and experiences need to be engaged with in response to the problem, and how this can open up different orientations towards the world (in terms of challenging assumptions and values). More specifically, emphasising the global aspect of this perspective, Principle 7 can stimulate a number of organising questions related to defining an inquiry problem,

- How do students relate their initial subjective inquiry interests to global existential concerns?
- How is the group inquiry interest related to global existential concerns, reflecting on whether their problem definition privileges the interest of the global North at the expense of the global South?
- When considering epistemologies and methodologies, do these reproduce the dominance of Northern epistemologies or do students enter into a critical dialogue with alternative epistemologies from both within the global North and global South?
- Do they orient their projects in relation to and in solidarity with social movements for justice and sustainability, and what are the consequences of orienting their projects in this way?

RUC's pedagogic model of PPL, and especially its principles of problem-orientation and exemplarity, have offered a potentially radical option in higher education. The addition of a seventh principle, that of international insight and vision, can enhance the radical and transformative possibilities of PPL. This appears even more important in light of the existential threats facing the world. This paper suggests that the potential of this new principle of internationalisation is best served by articulating it as a values-based pedagogy that emphasises global social justice.

This way of formulating internationalisation offers a means of translating it into pedagogical actions that enhance the radical potential of PPL. However, this also has application in all forms of problem-oriented pedagogies, by asking whose interests are served by the problems that students inquire into and what knowledge is deployed and produced. This paper will, therefore, conclude by elaborating on how the specific focus of discussion here articulates (a) the applicability of exemplarity to all forms of problem-oriented pedagogy, and (b) how this way of thinking is a reply to the call for pedagogic responses to climate change.

Although the pedagogic context for the theoretical exploration has been that of RUC, the concept of exemplarity has more general applicability. The exploration of pedagogy that explicitly seeks to formulate inquiry-problems as exemplary problems enables us to consider the usefulness of exemplarity as a pedagogic principle in all problem-oriented pedagogies. The challenge of exemplarity as world-orientation, as developed in this paper, provokes reflection on the way inquiry-problems can reflect the implicit hierarchies of knowledge and experience embedded in what Vanessa Andreotti and Lynette Shultz have referred to as Northern epistemologies [52,53]. This paper, then, is not inviting other problem-oriented pedagogies to emulate that of RUC. Instead, the invitation is to respond positively to Lynette Shultz' outline of a pedagogy based on global social justice. Related to this is the growing call for pedagogic responses to climate change and the ensuing climate crisis [60–62]. The paper began by describing possible scenarios—melting glaciers in Greenland, and migration—as examples where inter- or transdisciplinarity are required. These are also examples of exemplary issues where a world-oriented pedagogy is required. The theoretical reflections conducted here, and the option for a conception of world-orientation based on global social justice, are a response to this call. The exploration of a particular instance of problem-oriented pedagogy, RUC, acts as an exemplary case for theorising the nature of the inquiry-problem in dangerous times.

## 6. Conclusions

Central to this paper is a reconceptualization of the inquiry-problem in problem-oriented pedagogies. Specifically, the inquiry-problem is considered in relation to the exemplary nature of the problem, and of exemplarity understood as world-oriented. World-orientedness, as articulated here, calls for exemplary problems to hold the potential to awaken an interest in the world within students. The world, in this sense, is the world beyond the subjective interests of the student, one that opens up ethical and political relations with others in the world, and with the non-human world.

Although this conceptual exploration was stimulated by the introduction of a global citizenship and internationalisation strategy in one university, this reconceptualization is proposed as relevant to problem and inquiry-based pedagogies more widely. It is seen as reinvigorating the transformational potential of the problem-oriented pedagogy at RUC. More importantly, though, it is proposed that

such a reconceptualization of the inquiry-problem is a relevant pedagogic response to the existential threats facing the world.

**Funding:** This research received no external funding.

**Conflicts of Interest:** The author declare no conflict of interest.

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
