# Peer review of "Re-Thinking the “Problem” in Inquiry-Based Pedagogies through Exemplarity and World-Oriented"

_education, doi:10.3390/educsci9040295_

Round 1

Reviewer 1 Report

This paper tells a potentially interesting story that would be of interest to an international academic audience. However, the current text rather buries the key points that need to be brought forward from the background of the text. At the moment, the text is rather descriptive (reading a little like a policy report rather than a research paper). I feel it needs to have a more incisive, critical and analytic angle for it to be attractive to a wide readership. There are, I think, a few things that the authors could do to achieve this – some major, some very minor:

The title is a little misleading. Emphasis on ‘world’ (appearing twice in the title) and ‘global’ rather distracts the reader from the reality that we are considering a single-institution case study. As such I think that the title should be changed.

The Introduction curiously lacks punch – even though the authors have (understandably) offered consideration of some big ideas here. I get it – the implications are big, but again this is not the focus of the paper. The introduction comes across as a bit of a ramble through various ideas from melting glaciers in Greenland to Syrian migrants. There are also some assertions that may hold true for this institution, but are not necessarily generalizable- e.g. that programmes are unlikely to include political science and climatology. There also needs to be some consistency in terminology (and explanation of same – e.g. interdisciplinary [line 29] and transdisciplinary [line 35]. Overall, the introduction doesn’t draw the reader in with the appropriate expectations for the rest of the paper – a personal perspective, I know. But I have to say it as I see it. Finally, in this section, it seems a little late on page 2 [line 83] to say, “this paper begins…..”. This needs to come earlier. The authors need to think ‘what are the big ideas that we need to get across here’ and then place those front and centre to draw the reader in.

What is not immediately clear  is how this paper offers something new when compared with the book by Anderson and Heilesen? That work offers a comprehensive consideration of the Roskilde model. What are we adding here that will add value to the reader? The paper seems to focus on the addition of Principle 7. This could be emphasised with a graphic to summarise the model and to show how Principle 7 meshes with, and extends the model.

There are some hidden assumptions within the text that need to be more explicitly explained. E.g. line 146 – “educational process needs to transform that experience”. Could be criticised for offering a deficit focus on students.

Having explored the RUC experience, what are the implications that we could take away for our own institutions? This is rather important. Whilst the reader may well be interested in the RUC experience, many readers will want to know what they can learn from your experience and what ideas might they be able to take back (in whole or in part) to inform practice within their own university. If the authors just want to describe their institutional model, then this would not be appropriate for an international readership. For that, they need to broaden out the wider implications for pedagogy in Higher Education.

Some very minor points:

Care with apostrophes – e.g. line 145 should be “ students’ ”

What is the significance of the AACU [page 7], and how has this influenced practice in Denmark?

Some quotes in the text indicate the page in the citation [e.g. lines 306, 333, 390], but this is not true for the larger quotations. Is this the correct format?

The idea of threshold concepts seems to have been added as a bit of an after-thought (refs 50 and 51). It feels that this should either come much earlier as a frame for some of the discussion, or should be removed.

Check the references fully, e.g. in ref [36] Naidoo 2009, the title of the paper is incomplete. The references are all ‘double numbered’.

Author Response

Title and Abstract: the title and abstract have been edited to reflect the primary focus on exploring the nature of the inquiry problem. Introduction The introduction has been restructured so that the purpose of the paper as a theoretical and conceptual exploration has been foregrounded inter and trans-disciplinarity have been clarified (lines 50-51) I disagree with the judgement that the text on real-world problems such as the melting of glaciers on Greenland and migration is 'rambling' but I have moderated the assertion about differences between RUC and other institutions regarding subject combinations (lines 54-55).  In the conclusion I bring the conversation back to these examples The paper and the Andersen et al book are different entities and genres of text.  However, I have clarified the difference and the specific contribution of the paper (lines 206-209) I disagree that the argument (by Dewy and others) that educative problems need to have transformative potential for the student suggests a deficit view of students. The reviewer does not offer a more extended rationale for how this could be so.  I have added more discussion of what is meant by transformation to clarify this matter (lines 176-197) The reviewer's concern that the paper's focus on the RUC experience may be regarded as too local is, of course, an important point.  This has helped me to re-read the text and so I have enhanced the text to more explicitly state that the local example offered an exemplary case for exploring the nature of the inquiry-problem (in particular lines 484-507) AACU has been removed and replaced with revisiting RUC's Principle 7, allowing for a further critique of that All references have been checked and page numbers added where appropriate.  Citation and referencing style complies with the journal I do not really understand why the reference to threshold concepts is considered an 'after-thought'.  It was inserted to draw parallels between the concept of exemplarity as developed in German and Scandanavian educational thought and a relevant concept in Anglo-American educational thought.  It would be inappropriate to frame the discussion using threshold concepts when the framing is provided by Negt and Wagenschein.

Reviewer 2 Report

The article is well written and the topic is very interesting and important. The only problem I had with this paper was that there is not clear research questions, data or method. It makes it impossible to evaluate the study as a research report. Of course, all scientific articles do not have to be empirical but the authors talk about the Roskilde model which could be seen as a study case which could be analysed. In that case there should be clearly defined data. Or is the PPL the object of the study? What would be the data in that case? If the article aims to be purely theoretical then there should be more theoretical discussions and conceptual analysis for example about the concept of global learning. I think that the authors should decide whether they want to deepen their article to concentrate on studying Roskilde model/PPL method or on the theoretical discussions about global learning etc.

Author Response

I feel that the reviewer and I are coming from different disciplinary paradigms with different normative expectations.  However, I have tried to address the central concern with the primary focus of the paper.

I have therefore clarified matters in the Abstract, Introduction and conclusion that the paper constitutes a theoretical and conceptual exploration of the inquiry-problem as an exemplary problem, using the case of RUC as an exemplary case for examining the nature of the inquiry-problem in problem-oriented pedagogies more generally.  

Round 2

Reviewer 1 Report

the authors have made some progress in developing this paper, but I think there are still a few things to consider before publication.

The title has been amended, but still does not flow well and will not capture the attention of the readership. Use of the word ‘problem’ three times within the title is an indicator of this. The authors need to think about the key message of the paper and use this as the title.

There are still some clumsy expressions that need to be smoothed out. For example, in the revised abstract, you start, “The paper conducts .........”.   This leaves the researcher without a voice. The paper only reports a theoretical exploration that was undertaken by the researchers. This needs to be rephrased.

It may seem superficial, but by having a poor title and a contestable opening sentence in the abstract and introduction, the authors will have already lost the attention of many readers. The opening of the paper needs to invite the reader in, to want to read more. This is therefore something that requires a relatively small change, but will have a significant impact.

The introduction starts with the same phrase as the abstract.

Rather than ‘bachelor’s level’, I would say ‘undergraduate level,. - line 66.

Some corrections appear to repeat the original text .? See line 94 and 139, and 294-297, and 306. And so on. In consequence, the degree of editing is much less than it immediately appears.

The term ‘exemplarity’ (line 202) is not commonly used. I think a simpler term would be better here at times in the text to convey your meaning. certainly, I think the term needs to be better explained when it is first introduced, e.g line 190. A I understand it, exemplarity is similar to the ideas of Karl Maton, when he talks about weaving between high and low semantic gravity.

The paper needs a section 6 - a conclusion - to bring the key points together and synthesis the crux of the argument.

Author Response

Thank you for the helpful second review.

The title has been amended, but still does not flow well and will not capture the attention of the readership. Use of the word ‘problem’ three times within the title is an indicator of this. The authors need to think about the key message of the paper and use this as the title.   The title has been amended and hopefully now more closely fits the central claim of the paper

There are still some clumsy expressions that need to be smoothed out. For example, in the revised abstract, you start, “The paper conducts .........”.   This leaves the researcher without a voice. The paper only reports a theoretical exploration that was undertaken by the researchers. This needs to be rephrased. This has now been rephrased.

The introduction starts with the same phrase as the abstract. The beginning of the Introduced has been edited.

Rather than ‘bachelor’s level’, I would say ‘undergraduate level,. These have been changed

The term ‘exemplarity’ (line 202) is not commonly used. I think a simpler term would be better here at times in the text to convey your meaning I take note of the reviewer's comments here.  I have edited the Introduction in an attempt to address, in part, the reviewer's comments. I have not added an early definition of the term but have signposted where such a definition is provided (the first one being in section 2).  I hope that this meets with approval.

The paper needs a section 6 - a conclusion - to bring the key points together and synthesis the crux of the argument. Section 5 was originally conceived as a conclusion.  However, the addition of section 6 is, I think, an improvement.

Some corrections appear to repeat the original text .? See line 94 and 139, and 294-297, and 306.  I have not been able to respond to this comment.  The lines referred to do not appear to correspond with either the version I uploaded nor the version I downloaded.  Without further specification, I cannot address these at the moment.  Unless they are substantive in nature perhaps the reviewer will pass on these. Otherwise, I will address them if they are specified more clearly. 

Reviewer 2 Report

I am now satisfied with the changes made by the author. 

Author Response

Thank you for your reviews.

Round 3

Reviewer 1 Report

The authors have addressed all the comments and the paper is now ready for publication